# Describing, predicting and explaining adherence to total skin self-examination (TSSE) in people with melanoma: a 12-month longitudinal study

Julia L Allan [iD],[1] Derek W Johnston,[2] Marie Johnston,[1] Peter Murchie [iD] [3]

[1]Aberdeen Health Psychology Group, Institute of Applied Health Sciences, University of Aberdeen, Aberdeen, UK
[2]Aberdeen Health Psychology Group, School of Psychology, University of Aberdeen, Aberdeen, UK
[3]Centre for Academic Primary Care, Insitute of Applied Health Sciences, University of Aberdeen, Aberdeen, UK

**Correspondence to**
Dr Julia L Allan;
j.allan@abdn.ac.uk

## ABSTRACT

**Objectives** To describe trajectories in melanoma survivors' adherence to monthly total skin self-examination (TSSE) over 12 months, and to investigate whether adherence trajectories can be predicted from demographic, cognitive or emotional factors at baseline.

**Design** A longitudinal observational study nested within the intervention arm of the ASICA (Achieving Self-Directed Integrated Cancer Aftercare) randomised controlled trial.

**Setting** Follow-up secondary care in Aberdeen and Cambridge UK.

**Participants** n=104 adults (48 men/56 women; mean age 58.83 years, SD 13.47, range 28–85 years; mean Scottish Index of Multiple Deprivation score 8.03, SD 1.73, range 2–10) who had been treated for stage 0–IIC primary cutaneous melanoma in the preceding 60 months and were actively participating in the intervention arm of the ASICA trial.

**Interventions** All participants were using the ASICA intervention—a tablet-based intervention designed to support monthly TSSE.

**Primary and secondary outcome measures** The primary outcome was adherence to guideline recommended (monthly) TSSE over 12 months. This was determined from time-stamped TSSE data recorded by the ASICA intervention app.

**Results** Latent growth mixture models identified three TSSE adherence trajectories (adherent –41%; drop-off –35%; non-adherent –24%). People who were non-adherent were less likely to intend to perform TSSE as recommended, intending to do it more frequently (OR=0.21, 95% CI 0.06 to 0.81, p=0.023) and were more depressed (OR=1.31, 95% CI 1.06 to 1.61, p=0.011) than people who were adherent. People whose adherence dropped off over time had less well-developed action plans (OR=0.78, 95% CI 0.63 to 0.96, p=0.016) and lower self-efficacy about TSSE (OR=0.92, 95% CI 0.86 to 0.99, p=0.028) than people who were adherent.

**Conclusions** Adherence to monthly TSSE in people treated for melanoma can be differentiated into adherent, drop-off and non-adherent trajectories. Collecting information about intentions to engage in TSSE, depression, self-efficacy and/or action planning at outset may help to identify those who would benefit from additional intervention.

**Trial registration number** ClinicalTrials.gov Registry (NCT03328247).

## STRENGTHS AND LIMITATIONS OF THIS STUDY

⇒ A large sample of people treated for primary cutaneous melanoma were observed over 12 months to determine frequency and maintenance of adherence to recommended total skin self-examination (TSSE) over time.

⇒ TSSE adherence was objectively recorded via interactions with an app.

⇒ A three-class latent growth mixture model produced adherence classifications that were highly satisfactory statistically and interpretable.

⇒ The sample showed limited variance in socioeconomic status, tending to be more affluent than the general population.

⇒ Quality of TSSE (ie, thoroughness) was not a focus of this study and should be explored in future studies.

## INTRODUCTION

Melanoma is a relatively common skin cancer, with more than 16 000 new cases each year in the UK.[1] Post-diagnosis, around 20% of patients with melanoma will experience a recurrence[2] and 4%–8% will develop a new primary tumour,[3] most commonly within 5 years. As the majority of melanomas are readily visible on the skin's surface and show observable changes over time, clinical guidelines for the management of potential recurrence emphasise the importance of regular skin examination.[4 5]

Gold-standard total skin self-examination (TSSE) involves regular 'careful, deliberate and purposeful examination of the skin',[6] ideally monthly.[7] It has been estimated that regular TSSE can reduce mortality from melanoma by as much as 63%[6] as TSSE allows patients to detect recurrences/new primaries between scheduled follow-ups[8–12] when tumours are thinner/at an earlier stage.[13 14]

Despite the clear benefits of TSSE, adherence among people with melanoma is typically suboptimal. Surveys suggest that around 12% of patients never engage in recommended

TSSE[15] having never been directly advised to perform it[16] or advised to perform it but not shown how.[17] When TSSE is performed, it is often insufficiently frequent: a recent systematic review of 30 studies demonstrated that only a minority of patients with melanoma (16%–24%) adhere to recommendations by performing TSSE monthly.[18]

Interventions developed to support engagement in regular TSSE have demonstrated some positive results. For example, in one randomised controlled trial with over 1300 primary care patients, those randomised to an intervention that supported TSSE performed significantly more TSSEs at 2, 6 and 12 months than the control group.[19] However, other studies have suggested that the beneficial effects of TSSE interventions may dissipate over time.[20]

As sustained adherence to monthly TSSE is 'crucial' in melanoma care,[21] it has been suggested that regular reminders and booster educational sessions should be included in interventions to try and minimise drop-off in optimal TSSE over time.[21] The same authors report that self-efficacy (ie, confidence in ability to perform a TSSE) and intentions to perform a TSSE are two of the strongest psychosocial predictors of engagement in TSSE[22] and that as such, should be included in TSSE interventions. Specifically, they suggest that TSSE interventions should incorporate clear demonstrations of how to perform TSSE and concrete instructions about what to do if a suspicious change is detected (to enhance self-efficacy) and should include techniques such as implementation intentions (short specific, goal-focused plans[23]), to help people convert their intentions into actions.

One intervention which includes all of the content suggested to optimise TSSE over time is ASICA (Achieving Self-Directed Integrated Cancer Aftercare), a digital intervention designed to support high-quality, regular TSSE in people with cutaneous melanoma.[24 25] The details of intervention development are reported elsewhere[25] but in brief, ASICA is a tablet (Android)-delivered behavioural intervention based on the Information–Motivation–Behaviour model[26] and including behaviour change techniques based on Control Theory[27] and Implementation Intentions[23] to support engagement in TSSE. The ASICA intervention prompts and supports TSSE in people with melanoma by: (1) motivating people to perform TSSE; (2) building confidence (self-efficacy) in the ability to perform TSSE; (3) prompting monthly TSSE; (4) supporting thorough TSSE with body area guides; (5) prompting planning of TSSE and (6) prompting timely action if skin changes are detected.

ASICA was recently evaluated in a randomised controlled trial over 12 months with 240 patients with melanoma[28] to determine the intervention's effect on TSSE behaviour and psychological well-being. One objective of the trial was to collect detailed data on the frequency and maintenance of TSSE behaviour over time in order to investigate whether clinically relevant patterns in adherence behaviour can be detected and predicted early on, and whether this knowledge can be used to optimise future interventions.

The present study aims to answer two research questions:
1. What does adherence to TSSE over 12 months look like? Are there distinct patterns of non-adherence?
2. What factors predict suboptimal TSSE adherence? That is, do groups with suboptimal adherence differ from those with optimal adherence on demographic (age, gender, socioeconomic status, rurality), sociocognitive (intentions, self-efficacy, action plans) and/or emotional (recurrence worries, anxiety, depression) factors?

## METHODS
### Design
A longitudinal study of objectively recorded TSSE adherence in people with melanoma over 12 months. The study was embedded within the intervention arm of the recently completed ASICA randomised controlled trial.[25 28]

### Patient and public involvement
A detailed pilot study was conducted during the development of the ASICA project to ascertain patients' priorities, experiences and preferences. Interviews were carried out with 19 potential recipients of the ASICA intervention, and these interviews informed the development of the study research questions and selection of outcome measures. Patients were not directly involved in the design of the study but did inform the design via participation in the pilot study interviews. The burden of the ASICA intervention was assessed by patients in a qualitative substudy. Two patient representatives sat on the trial steering group, feeding into plans for recruitment and dissemination. The results of the project will be disseminated to all participants (other than those who opted out) via a postal newsletter.

### ASICA trial and intervention
The ASICA trial was a two-arm, multicentre randomised controlled trial designed to test the effects of the ASICA intervention on TSSE frequency and psychological well-being in adults treated for a stage 0–IIC[29] primary cutaneous melanoma. Participants were randomly allocated on a 1:1 basis to the ASICA intervention (intervention plus usual care) or control (usual care only) group. Intervention participants received the app-based ASICA intervention. The app included information about the importance of monthly TSSE, instructional videos demonstrating TSSE, a digital map of the patient's own skin, a checkbox list of body parts to check, prompts to plan TSSE and the capability to send photographs of suspicious skin lesions to a dermatology nurse practitioner for review. As part of the intervention, participants were prompted monthly (ie, in line with guideline frequency) to conduct a TSSE.

## Participants

Participants were n=104 adults (48 men/56 women; mean age 58.83 years, SD 13.47, range 28–85 years; mean Scottish Index of Multiple Deprivation (SIMD) score 8.03, SD 1.73, range 2–10) who had been treated for stage 0–IIC primary cutaneous melanoma in the preceding 60 months and who were actively engaged in the intervention arm of the ASICA trial between 2018 and 2020. Sample size for the ASICA trial (n=240 total; n=120 intervention group) was pragmatically determined based on the need to capture anticipated diversity in the primary outcomes (anxiety, depression, worry and quality of life) and the number of participants it was possible to recruit in the available time.[25]

## Measures

### TSSE adherence

TSSE adherence was determined from data collected by the ASICA intervention app. After completing a practice, participants were prompted every 30 days for the next 12 months to complete and report a TSSE. Adherence to TSSE guidelines was determined by assessing via the time-stamped data collected by the ASICA app whether participants had completed at least one TSSE during each 30-day period following the receipt of a prompt (ie, had engaged in TSSE at least monthly). Each period was coded 1 (at least one report received) or 0 (no report received), with possible scores ranging from 0 (totally non-adherent) to 13 (totally adherent, 1 practice+12 monthly TSSEs).

Demographic, cognitive and emotional factors were measured in a baseline questionnaire administered at the start of the ASICA trial.

### Demographic factors

Age in years and gender were self-reported. Socioeconomic status was coded 1 (most deprived) to 10 (least deprived) by linking postcodes to indices of deprivation (SIMD; English indices of deprivation).[30 31] Rurality/urbanity was coded from 1 (most urban) to 10 (most rural).[32 33]

### Cognitive factors

*Intention* towards engaging in TSSE was assessed with the item 'In the next 12 months, do you intend to check your skin for early signs of skin cancer? (yes/no)… If yes, how many times? (insert number)'. In order to map intentions onto guideline behaviour, this item was scored categorically as <12 (intention below guideline), =12 (guideline intention) or >12 (intention above guideline).

*Self-efficacy* towards TSSE was assessed with four items rated on a 10-point scale from 'not at all confident' to 'highly confident': 'How confident are you that… you can check your own skin correctly?; that you will find time in the next 12 months to check your own skin?; that you will remember to check your own skin at least once a month?; that if you find a spot or mole of concern that you will take appropriate action?'

*Action planning* about TSSE was assessed with two items rated on a 5-point scale from 'strongly agree to 'strongly disagree': 'I have made plans about when to examine my own skin' and 'I have made plans about where I will be when I examine my own skin'.

All of the items measuring cognitive factors were developed for the ASICA study (to relate specifically to the TSSE context) but were based on standard and valid methods of assessing the relevant theoretical constructs.[34–36]

### Emotional factors

*Anxiety* and *depression* were measured using the 14-item Hospital Anxiety and Depression Scale,[37] a valid measure with adequate internal consistency.[38] Anxiety and depression were scored separately (from 0 to 4), with higher scores indicating greater distress.

*Melanoma-specific worries* were assessed with the four-item Melanoma Worry Scale.[39] Items were rated on a 5-point scale from 'not at all' to 'almost all the time' and assessed the frequency and impact of worries about possible recurrence.

## Analysis

Latent group mixture modelling (LGMM) was carried out with Mplus V.8.4[40] to establish if there were systematically different classes of TSSE over the 12-month measurement period. A series of models were estimated from the TSSE adherence data with one to five latent classes, and were compared on the Bayesian Information Criterion (BIC), entropy values and number of participants per class. Class membership was obtained solely on the basis of the adherence data (ie, covariates were not included in the modelling exercise), and the resulting classification information was then combined with the other information to explore the differences between the classes using the classic three-step method.[41] In the present analysis, the LGMM identified three classes of adherence behaviour (adherent, drop-off and non-adherent) by modelling the proportion of people who had completed the recommended monthly TSSE within each successive 30-day period over 12 months. Higher proportional estimates in this modelling exercise mean that more people are completing the recommended TSSE within each time window. Logistic regressions were then conducted using SPSS V.25 to predict membership of these three classes from demographic (age, gender, socioeconomic status, rurality), cognitive (intentions, self-efficacy and action plans about TSSE) and emotional (anxiety, depression and melanoma worry) factors. In these logistic regression analyses, the dependent variable was dummy coded (with the adherent coded as 0 and the non-adherent or drop-off coded as 1). Categorical intention scores were treated as unordered categories. Missing data, which were <4% on all measures other than intention and 11.5% for intention, were dealt with by item deletion. Intention analyses were rerun with missing values imputed and results did not change.

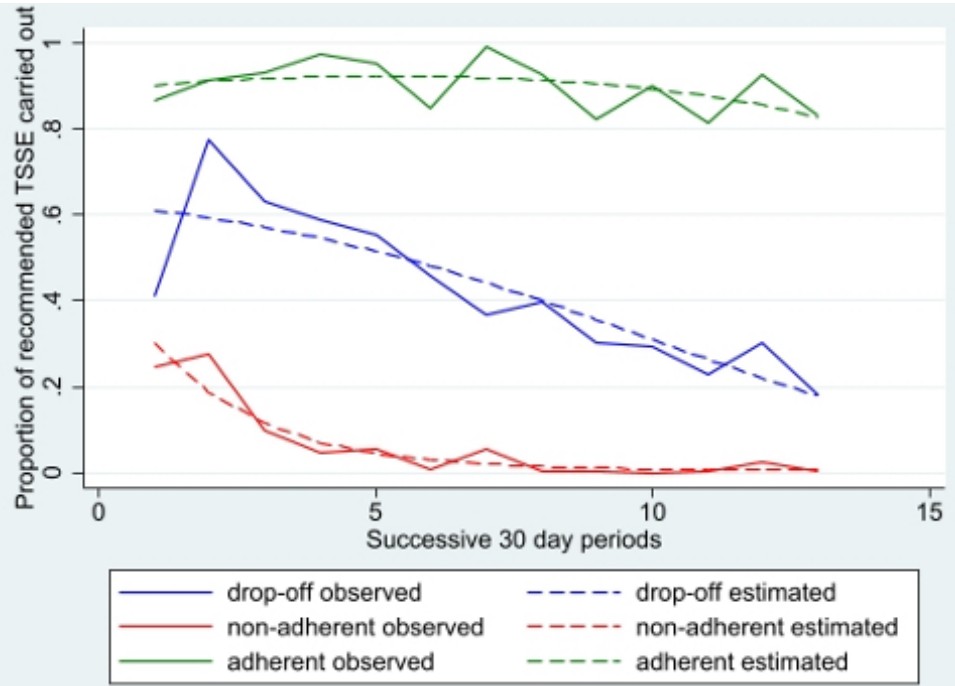

**Figure 1** The three classes of adherence to TSSE over time showing LGMM estimated (statistically predicted) and observed (actual) proportion of participants completing TSSE within each 30-day period. LGMM, latent group mixture modelling; TSSE, total skin self-examination.

## RESULTS

**What does adherence to TSSE over 12 months look like? Are there distinct patterns of non-adherence?**

Of the 104 participants, 12 did not submit any TSSE reports using ASICA and the remainder submitted between 1 and 13 (median 6.5, IQR 9.75).

LGMM was tested with one to five latent groups. The lowest (best) BIC was for a three-class solution (1318). The entropy score (a measure of the quality of the classification) for the three-class solution was also highly satisfactory (0.906). The number of participants per class was substantial with the best fitting model classifying reasonable numbers (n=25/n=36/n=43) into categories that were readily interpretable (as showing consistent non-adherence; adherence which dropped off over time and consistent adherence, respectively). The three-class solution was therefore selected for further study.

Figure 1 shows the LGMM estimated and observed values plotted over time and illustrates the three distinct classes identified. The largest class are people who are adherent (n=43) who achieve over 90% adherence across the study period and show no significant decrease over time (p=0.188). The next class, people whose adherence dropped off over time (n=36), are those who start well, particularly after the first prompt but whose TSSE adherence declines steadily over 12 months (p<0.001) until only around 20% are providing the required reports by the final month. The final group, people who are non-adherent (n=25) start poorly, submitting only 28% of reports in the first month, and decrease rapidly such that they are providing virtually no reports by 6 months (p=0.022). An overview of the characteristics

of participants in the three TSSE adherence classes can be found in table 1. The accuracy of the classification (likely vs actual classification) was high (adherent=0.98; drop-off=0.93; non-adherent=0.96).

**Why does optimal and suboptimal TSSE adherence occur? That it is, do groups with suboptimal adherence differ from those with optimal adherence on demographic (age, gender, socioeconomic status, rurality) sociocognitive (intentions, self-efficacy, action plans) and/or emotional (recurrence worries, anxiety, depression) factors?**

The results of the univariate logistic regression analyses predicting adherence class (adherent; drop-off; non-adherent) from the demographic, cognitive and emotional measures are shown in table 2 (comparison of non-adherent and adherent) and table 3 (comparison of drop-offs and adherent). Supplemental multivariate analyses can be found in online supplemental tables 1 and 2.

Age, sex and SIMD did not significantly predict adherence class in any analysis, indicating that demographic factors alone are not sufficient to predict different adherence trajectories over time. However, there was a tendency for rural participants to be more adherent.

In univariate comparisons of *cognitive factors*, people who were non-adherent were less likely to intend to carry out TSSE monthly than people who were adherent, instead reporting (potentially unrealistic) intentions to perform TSSE more often than monthly. People who were non-adherent did not differ from people who were adherent in terms of action plans or self-efficacy towards TSSE. People whose adherence dropped off did not differ significantly from people who were adherent in

**Table 1** Demographic, cognitive and emotional factors in the three adherence groups

|  | Adherent | Drop-off | Non-adherent |
|---|---|---|---|
| N | 43 | 36 | 25 |
| Age in years (M, SD) | 60.45 (11.77) | 57.12 (13.31) | 58.50 (16.38) |
| Gender (M/F) | 16/27 | 20/16 | 12/13 |
| Rurality (urban/rural) | 21/22 | 23/13 | 18/7 |
| Social deprivation (M, SD) | 7.98 (1.50) | 8.17 (1.72) | 7.92 (2.14) |
| Action planning (M, SD) | 6.86 (2.10) | 5.56 (2.47) | 6.25 (2.42) |
| Self-efficacy (M, SD) | 32.32 (5.89) | 28.85 (7.04) | 30.50 (6.90) |
| Intentions (n)* |  |  |  |
| <12 times | 13 | 15 | 6 |
| =12 times | 22 | 11 | 7 |
| >12 times | 6 | 3 | 9 |
| HADS anxiety (mean, SD) | 4.23 (3.99) | 5.33 (3.84) | 6.13 (4.95) |
| HADS depression (mean, SD) | 1.93 (2.19) | 3.14 (3.45) | 3.75 (3.05) |
| Melanoma worry (mean, SD) | 8.33 (3.52) | 8.73 (3.59) | 8.75 (3.90) |

*Only 92 of 104 participants completed the intention measure.
HADS, Hospital Anxiety and Depression Scale.

their TSSE intentions. However, the people whose adherence dropped off reported lower levels of action planning and self-efficacy than people who were adherent. When *emotional factors* were compared, people who were non-adherent had significantly higher depression scores than people who were adherent with people whose adherence dropped off intermediate but not reliably different from people who were adherent. Anxiety was slightly higher in people who were non-adherent but this was not significant at p<0.05. Melanoma worries did not significantly differ between any of the adherence groups.

## DISCUSSION

Distinct patterns in TSSE adherence emerged in participants using the ASICA intervention over 12 months. Of the 104 participants, 41% (those in the 'adherent' category) consistently completed >90% of guideline-recommended TSSEs over 12 months, which compares favourably with the 16%–24% of participants reporting optimal TSSE adherence in a recent systematic review.[18] However, even with the theory and evidence-based support offered by ASICA, the remaining 59% of participants in

**Table 2** Logistic regression analyses of demographic, cognitive and emotional factors in people who are non-adherent (comparison group) versus adherent

|  | B | SE | Wald | P value | OR | 95% CI |
|---|---|---|---|---|---|---|
| **Univariate** |  |  |  |  |  |  |
| Age | 0.01 | 0.02 | 0.33 | 0.566 | 0.99 | 0.95 to 1.03 |
| Gender | 0.44 | 0.51 | 0.76 | 0.385 | 1.56 | 0.57 to 4.23 |
| SIMD | −0.02 | 0.14 | 0.02 | 0.9 | 0.98 | 0.74 to 1.30 |
| Rurality | −0.99 | 0.54 | 3.37 | 0.066 | 0.37 | 0.13 to 1.07 |
| Intention <12* | −1.18 | 0.72 | 2.67 | 0.103 | 0.31 | 0.08 to 1.27 |
| Intention=12* | −1.55 | 0.68 | 5.17 | 0.023† | 0.21 | 0.06 to 0.81 |
| Action plan | −0.13 | 0.12 | 1.17 | 0.28 | 0.88 | 0.70 to 1.11 |
| Self-efficacy | −0.04 | 0.05 | 1.18 | 0.28 | 0.96 | 0.88 to 1.04 |
| HADS anxiety | 0.1 | 0.06 | 2.76 | 0.096 | 1.10 | 0.98 to 1.24 |
| HADS depression | 0.27 | 0.11 | 6.43 | 0.011† | 1.31 | 1.06 to 1.61 |
| MWS | 0.04 | 0.07 | 0.23 | 0.633 | 1.04 | 0.90 to 1.19 |

*Intention >12 used as comparison.
†Significant at p<0.05.
HADS, Hospital Anxiety and Depression Scale; MWS, Melanoma Worry Scale; SIMD, Scottish Index of Multiple Deprivation.

**Table 3** Logistic regression analyses of demographic, cognitive and emotional factors in people whose adherence dropped off over time (comparison group) versus people who are adherent

|  | B | SE | Wald | P value | OR | 95% CI |
|---|---|---|---|---|---|---|
| **Univariate** | | | | | | |
| Age | −0.02 | 0.02 | 1.38 | 0.241 | 0.98 | 0.94 to 1.02 |
| Gender | 0.75 | 0.46 | 2.63 | 0.105 | 2.11 | 0.86 to 5.2 |
| SIMD | −0.08 | 0.14 | 0.28 | 0.597 | 1.08 | 0.81 to 1.43 |
| Rurality | −0.62 | (0.46) | 1.78 | 0.182 | 0.54 | 0.22 to 1.33 |
| Intention <12* | 0.84 | 0.8 | 1.09 | 0.297 | 2.31 | 0.48 to 11.12 |
| Intention=12* | 0 | 0.8 | 0 | 1 | 1 | 0.21 to 4.78 |
| Action plan | −0.25 | 0.11 | 5.77 | 0.016† | 0.78 | 0.63 to 0.96 |
| Self-efficacy | −0.08 | 0.04 | 4.82 | 0.028† | 0.92 | 0.86 to 0.99 |
| HADS anxiety | 0.07 | 0.06 | 1.52 | 0.217 | 1.08 | 0.96 to 1.21 |
| HADS depression | 0.16 | 0.09 | 3.19 | 0.074 | 1.17 | 0.98 to 1.40 |
| MWS | 0.04 | 0.07 | 0.26 | 0.61 | 1.06 | 0.91 to 1.18 |

*Intention >12 used as comparison.
†Significant at p<0.05.
HADS, Hospital Anxiety and Depression Scale; MWS, Melanoma Worry Scale; SIMD, Scottish Index of Multiple Deprivation.

the present study showed one of two patterns of suboptimal adherence. Approximately 35% of people (those in the 'drop-off' category) initially adhered well but showed steady and significant declines in TSSE adherence over time until only around one-fifth were conducting TSSEs by 12 months. The remaining 24% (those in the 'non-adherent' category) consistently failed to engage in monthly TSSEs, with only a minority (around one-third) conducting a TSSE in the first month and virtually none completing one from 6 months onwards.

## Predictors of adherence

One of the aims of the present study was to determine whether participants who would go on to display suboptimal adherence could be identified at baseline so that they could be better supported from intervention outset. To this end, adherence trajectories were predicted from a range of easily measurable and theoretically relevant demographic, cognitive and emotional factors. None of the demographic factors assessed in the current study (age, gender, socioeconomic status, rurality) differentiated between classes of TSSE adherence behaviour, although there was a slight tendency for those living in urban areas to be less adherent. It should be noted, however, that participants in this study showed extremely limited variance in socioeconomic status, tending to be more affluent than the general population. This restricted range likely reflects the combined effects of melanoma incidence being higher in more affluent individuals (eg, 42), and trial participation being skewed towards individuals from higher socioeconomic groups, and limits the conclusions that can be drawn about socioeconomic status.

When cognitive factors (intentions about TSSE, self-efficacy towards TSSE and plans about when and where

to perform TSSE) were examined, the data suggested that different cognitive processes may be involved in the two suboptimal patterns of adherence observed ('non-adherent' and 'drop-off'). People who were non-adherent were significantly more likely to have high and potentially unrealistic intentions about TSSE than people who were adherent: with many intending to perform TSSEs more frequently than monthly. Neither TSSE self-efficacy nor action planning significantly differentiated people who were non-adherent from people who were adherent. In contrast, the opposite pattern was found to differentiate the people whose adherence dropped off from people who were adherent: the two groups did not significantly differ in their intentions about TSSE but people whose adherence dropped off showed significantly lower levels of action planning and self-efficacy than people who were adherent. The observed pattern of results fits well with theoretical models of health behaviour change such as the Health Action Process Approach (HAPA[43]). HAPA explicitly distinguishes between motivational processes (such as intentions) involved in the early goal setting phase of behaviour change, and volitional processes (such as self-efficacy and planning) involved in the later goal pursuit phase of behaviour change. In the present study, the participants who were least likely to 'get started' (ie, the people who were non-adherent) showed measurable differences in motivational processes (intentions) relative to people who were adherent. Similarly, those who started well (had motivation) but then had issues with maintaining the volitional aspects of regular TSSE over time (ie, those whose adherence dropped off) showed lower levels of self-efficacy and planning than those who were adherent. Cognitive factors were on average less predictive of TSSE behaviour in the present study than in other

melanoma studies,[22] possibly because the ASICA intervention was designed to boost intentions, self-efficacy and planning. This may have reduced natural variance in these constructs, limiting their predictive power.

The analysis of the emotional factors (anxiety, depression and melanoma-specific worries) indicated that while people who were adherent consistently had the 'best' scores (ie, lowest anxiety, lowest depression, fewest worries about recurrence), the only factor that significantly differentiated people who were adherent from people who were non-adherent was depression. People whose adherence dropped off, in contrast, did not significantly differ from people who were adherent on any of the emotional factors. It is noteworthy that melanoma-specific worries were remarkably consistent across adherence groups and did not predict TSSE behaviour. Fears about recurrence are often suggested as a likely predictor of TSSE behaviour (ie, that people worried about recurrence will check their skin more frequently, potentially more frequently than recommended), and recent evidence suggests that people who are more worried about recurrence are more likely to conduct TSSE and to conduct more thorough TSSE.[44] The present results were not in line with this, suggesting that recurrence fears were unrelated to TSSE adherence over 12 months.

### Clinical implications
The present results suggest that routinely screening for fear of recurrence will not helpfully discriminate patients with melanoma in terms of their likely future TSSE adherence. Making baseline assessments of participants' intentions about TSSE and level of depression however may serve to highlight the group most likely to be non-adherent. Intentions, if suboptimal, can in theory be strengthened with three behaviour change techniques: giving information about the health consequences of TSSE, setting goals about completing TSSE and providing an incentive for reaching the outcome.[45] Similarly, depression might be reduced by helping patients to more effectively manage negative emotions, for example, through stress management techniques.[45] It is important to note however that the effect sizes in the present study were small so the potential clinical benefit of this approach would be limited.

Identifying those who start well but falter (those whose adherence 'drops off') is challenging. While measures of self-efficacy and action planning differentiated people whose adherence dropped off from people who were adherent on average, they would not reliably identify people on this adherence trajectory at baseline (eg, baseline self-efficacy scores in the present data set would correctly identify only 46% of drop-offs). Instead, adherence behaviour would have to be monitored for 6 months (in the present data), to identify 75% of people whose adherence dropped off (as those who have completed <4 of the recommended 6 TSSEs). Top-up interventions could be targeted at this point: action planning might be addressed by prompting people to make 'if–then' plans[23]

and self-efficacy might be enhanced by methods such as verbal persuasion and focusing on past success (eg, in adhering in the early stages).[45] If top-up interventions were to be used, it would be possible in the present data to accurately identify 92% of people who were non-adherent and 100% of people who were adherent at 3 months on the basis of TSSEs completed (0–1 vs 2–3).

### Strengths and limitations
The present study has several methodological strengths. It uses objectively recorded TSSE behaviour rather than self-reports, and we can therefore be confident that TSSE took place at the times indicated. The study also followed participants over a full 12 months and tested both emotional and sociocognitive models of behaviour. However, as discussed above, one notable limitation of the current study is that participants came almost exclusively from the less deprived end of the socioeconomic spectrum. While broadly representative of the clinical population (melanoma being more common in high socioeconomic status individuals), the results reported are not representative of the general population. This raises an important point about whether digital interventions like ASICA would exacerbate existing health inequalities if uptake (and therefore potential to benefit) is higher in individuals from less deprived sectors of the population (cf. 46 47), or whether they would usefully 'free up' face-to-face resources for those least likely to engage with remote healthcare.

### Future directions
Future studies should investigate in more detail methods for early detection of those least likely to be adherent. Based on our data, interventions which boost action planning and self-efficacy at 3 and 6 months might be beneficial in reducing the number of people whose adherence drops off while an intervention prior to starting monthly TSSE to address motivation and depression might reduce the number of people who go on to be non-adherent. Detection of suboptimal adherence on an individual level would require real-time measurement and more dynamic and individualised assessments in order to model the thresholds beyond which optimal adherence can be reliably distinguished from adherence which is likely to decline in future. Future studies should also extend the present work on the frequency and maintenance of TSSE to TSSE thoroughness. Studies which have investigated the thoroughness of TSSE suggest that only 7%–14% of patients check their whole body while performing TSSE[44 48] and only 9%–13% use a mirror or ask another person to help them check areas that are difficult to see.[48] Melanoma can recur anywhere on the body and more thorough SSE is associated with detection of earlier stage, more survivable lesions.[49]

In conclusion, the present study demonstrates that adherence to recommended monthly TSSE in individuals treated for melanoma can be reliably differentiated into three distinct trajectories over time: adherent, drop-off

and non-adherent. Collecting information about intentions to engage in TSSE, depression level, self-efficacy and/or action planning at outset may help to distinguish between those who go on to engage in optimal, declining or consistently suboptimal adherence.

**Contributors** JLA, MJ, DJ and PM conceived of, designed and planned the study. DJ analysed the data in consultation with JLA, MJ and PM. JLA drafted the manuscript. JLA, DJ, MJ and PM interpreted the results and revised the manuscript. JLA is responsible for the overall content and acts as guarantor.

**Funding** This work was supported by a grant from a Cancer Research UK Population Research Committee project award (C10673/A21685).

**Disclaimer** The views and opinions expressed herein are those of the authors and do not necessarily reflect those of Cancer Research UK. The funder (through their peer review and funding board review process) approved the study proposal but had no role in the collection, analysis, or interpretation of data, or writing of the report.

**Competing interests** None declared.

**Patient and public involvement** Patients and/or the public were involved in the design, or conduct, or reporting, or dissemination plans of this research. Refer to the Methods section for further details.

**Patient consent for publication** Not required.

**Ethics approval** This study involves human participants and was approved by the North of Scotland Research Ethics Committee (17/NS/0040).

**Provenance and peer review** Not commissioned; externally peer reviewed.

**Data availability statement** Data are available upon reasonable request.

**ORCID iDs**
Julia L Allan http://orcid.org/0000-0001-7287-8363
Peter Murchie http://orcid.org/0000-0001-9968-5991

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
