## [Reviewer comments · BMJ Open]

ARTICLE DETAILS

TITLE (PROVISIONAL)	Describing, predicting and explaining adherence to Total Skin Self-Examination (TSSE) in people with melanoma: a 12 month longitudinal study
AUTHORS	Allan, Julia; Johnston, D; Johnston, Marie; Murchie, Peter

VERSION 1 – REVIEW

REVIEWER	Heckman, Carolyn Rutgers Cancer Institute of New Jersey, Division of Medicine
REVIEW RETURNED	19-Jan-2022

GENERAL COMMENTS	This paper is well-written and contributes to the field. Suggestions for improvement are provided below. Abstract: Reliability was not assessed. Intro: I suggest deleting the first question, which is difficult to assess, and just sticking with the second question: 1. Why does optimal and suboptimal TSSE adherence occur? 2. That it is, can TSSE adherence be predicted from socio-cognitive (intentions, self-efficacy, action plans) and/or emotional (recurrence worries, anxiety, depression) factors? Methods: Provide more information about what the estimated values are and mean. Are they necessary to present in Figure 1? Results Would you want to present significant differences among the three groups via univariate analyses in Table 1? I'm unclear why all three groups weren't presented together for multivariate analyses and also why the comparison between nonadherent and dropout wasn't presented. I would think that could also help determine differences in factors that might affect initiating vs. maintaining the behavior. Please use person-first language throughout, i.e., people who were nonadherent vs. non-adherents. Figure 2: This was too small for me to read. I'm also not sure what the importance is since the graphs seem very similar within groups. It seems the information is mostly illustrated already in Figure 1. Other: Please define SIMD at the first occurrence of the abbreviation.
---

REVIEWER	Vestergaard, Tine University of Southern Denmark
REVIEW RETURNED	31-Jan-2022

GENERAL COMMENTS	Thank you for an interesting and very well written paper. I have the following comments: The participants were treated for stage 0-IIC melanoma within the
--

	last 60 months. Did you look for a difference in TSSE-adherence between newly treated patients versus those treated 4 or 5 years ago? Did you look for a difference between melanoma stage? If not, do you think this might influence the adherence to monthly TSSE? How did you validate the items used to assess cognitive factors? Page 8 line 29: SIMD not defined. In the results section I do not understand the number of participants per class (25/36/34)? Reference 31 states "In preparation". The editor will have to decide whether that is acceptable.
--	---

REVIEWER	Bidoli, Ettore IRCCS CRO AVIANO, Epidemiology
REVIEW RETURNED	04-Feb-2022

GENERAL COMMENTS	This paper described and tried to explain adherence to TSSE in patients with melanoma. Although interesting, this paper needs some improvements: Major point: Please report in Methods section a power and sample size analysis. I'm afraid that some results strongly depend on the power of the study. Please report in Results section the ORs of "Intention" and "HADS depression" mutually adjusted and comment the results. Minor points: The Introduction section is too long and many parts are not informative with regards to the aim of the study. The "predictors of adherence" paragraph in Discussion section should be shortened in order to avoid redundancy. I am not sure about usefulness of Figure 2; it may be replaced by a sentence in Results section describing the overall observed pattern.
---

VERSION 1 – AUTHOR RESPONSE

Reviewer	Comment	Response
R1 (CH) 1	This paper is well-written and contributes to the field. Suggestions for improvement are provided below.	Thank you for your positive feedback. Responses to your suggestions for improvement are outlined below.
R1 (CH) 2	Abstract: Reliability was not assessed.	The word 'reliably' has been removed from the abstract to avoid the implication that reliability was formally assessed.
R1 (CH) 3	Intro: I suggest deleting the first question, which is difficult to assess, and just sticking with the second question: 1. Why does optimal and suboptimal TSSE adherence occur? 2. That it is, can TSSE adherence be predicted from socio-cognitive	We were somewhat unsure how to interpret this suggestion as there are 3 research questions, and the suggested wording appears to retain only the third. The 1st research question relates specifically to the main LGMM

	(intentions, self-efficacy, action plans) and/or emotional (recurrence worries, anxiety, depression) factors?	analysis and as such we are keen to retain it. However, we have reworded the 1st question to more clearly align with the analysis conducted and have combined the 2nd and 3rd research questions along the lines suggested (incorporating the demographic analysis into this combined question). The research questions are now given as;  1. What does adherence to TSSE over 12 months look like? Are there distinct patterns of non-adherence? 2. Why does optimal and suboptimal TSSE adherence occur? That is, do groups with suboptimal adherence differ from those with optimal adherence on demographic (age, gender, socioeconomic status, rurality), socio-cognitive (intentions, self-efficacy, action plans) and/or emotional (recurrence worries, anxiety, depression) factors?
R1 (CH) 4	Methods: Provide more information about what the estimated values are and mean. Are they necessary to present in Figure 1?	The analysis section has now been amended to make clear what the estimates modelled in the LGMM are and what these scores mean. The section now reads; “In the present analysis, the LGMM identified 3 classes of adherence behaviour (adherent, drop-off and non-adherent) by modelling the proportion of people who had completed the recommended monthly TSSE within each successive 30 day period over 12 months. Higher proportional estimates in this modelling exercise mean that more people are completing the recommended TSSE within each time window.” The title of Figure 1 has also been amended to make the distinction between the observed and estimated values clear; “Figure 1: The 3 classes of adherence to TSSE over time showing LGMM estimated (statistically predicted) and observed (actual) proportion of participants completing TSSE within each 30 day period.”
R1 (CH)	Results	As Table 1 gives descriptive summaries

5	Would you want to present significant differences among the three groups via univariate analyses in Table 1?	of different factors (means and frequencies), and the univariate analyses relate to the factors that predict category membership, we would prefer to keep the two separate to avoid giving the impression that the univariate analysis results represent differences in group means.
R1 (CH) 6	I'm unclear why all three groups weren't presented together for multivariate analyses and also why the comparison between nonadherent and dropout wasn't presented. I would think that could also help determine differences in factors that might affect initiating vs. maintaining the behavior.	The aim of the present study was to identify factors that predict suboptimal adherence (relative to optimal adherence), as our clinical and theoretical questions relate to deviation from guideline recommendations. We did not therefore directly compare the characteristics of those showing different types of suboptimal adherence. We have added some additional multivariate analyses (see response to Reviewer 3, point 3), however these aim to simultaneously include and adjust for the effects of the multiple predictors and do not simultaneously compare the 3 adherence groups.
R1 (CH) 7	Please use person-first language throughout, i.e., people who were nonadherent vs. non-adherents.	The labels used were adopted to be as concise as possible given the journal word limits but we agree that language should be person centred. This has been amended throughout the manuscript whenever we talk about people. For brevity, where we talk only about adherence categories in an analytic sense, we have retained the shorter labels. In sections with set word limits (e.g. the abstract), we have made other small changes to the text to accommodate the increase in length.
R1 (CH) 8	Figure 2: This was too small for me to read. I'm also not sure what the importance is since the graphs seem very similar within groups. It seems the information is mostly illustrated already in Figure 1.	The individual level graphs in Figure 2 were included as they illustrate the high level of individual consistency within each adherence group. However, we agree that they provide supplementary information and have therefore removed Figure 2 from the manuscript.
R1 (CH) 9	Other: Please define SIMD at the first occurrence of the abbreviation.	SIMD has now been defined on first use.
R2 (TV) 1	Thank you for an interesting and very well written paper.	Thank you for your positive feedback.
R2 (TV) 2	The participants were treated for stage 0-IIc melanoma within the last 60 months. Did you look for a	We did not look at the influence of clinical factors in the current study as our focus was on the cognitive

	difference in TSSE-adherence between newly treated patients versus those treated 4 or 5 years ago? Did you look for a difference between melanoma stage? If not, do you think this might influence the adherence to monthly TSSE?	and emotional measures that are not currently routinely collected in melanoma care but which are theoretically likely to be relevant to behaviours such as adherence. Participants were recruited from a 60 month window post-diagnosis (with no stratification of actual time since diagnosis) as recurrence risk is highest within this time period and TSSE is equally recommended to all. Within the present sample, the mean time since diagnosis was 2 years (SD=1.3 years), and the majority (83%) had stage 0/1 melanoma so it is unlikely that we would have sufficient variance to conduct meaningful analyses with these factors. We agree that clinical factors may influence adherence, but there is limited evidence in the literature to support this. For example, studies which have assessed the predictive power of factors such as tumour depth, location and family history of cancer find no significant impact of these factors on skin checking behaviour (Mujumdar et al, 2009). There is some evidence that people with cancers at a more advanced stage may be more likely to be thorough during TSSE (e.g. by being more likely to use a picture guide during and to ask another person to assist) but stage does not appear to predict whether or not people engage in TSSE in the first place (Körner et al, 2013). Mujumdar UJ, Hay JL, Monroe-Hinds YC, Hummer AJ, Begg CB, Wilcox HB, Oliveria SA, Berwick M. Sun protection and skin self-examination in melanoma survivors. Psychooncology. 2009 Oct;18(10):1106-15. Körner A, Coroiu A, Martins C, Wang B. Predictors of skin self-examination before and after a melanoma diagnosis: the role of medical advice and patient's level of education. Int Arch Med. 2013 Feb 27;6(1):8.
R2 (TV) 3	How did you validate the items used to assess cognitive factors?	We did not conduct any direct psychometric validation work within this study. However, the items used to measure cognitive factors are based on standard and valid methods of measuring each theoretical construct. A statement to this effect with supporting references has been added to the relevant section of the methods;

		"All of the items measuring cognitive factors were developed for the ASICA study (to relate specifically to the TSSE context) but were based on standard and valid methods of assessing the relevant theoretical constructs (34,35,36)."
R2 (TV) 4	Page 8 line 29: SIMD not defined.	SIMD has now been defined.
R2 (TV) 5	In the results section I do not understand the number of participants per class (25/36/34)?	Apologies, there is a typo in these numbers which should read (25/36/43), thank you for highlighting. This has now been corrected. These numbers are the results of the modelling procedure and demonstrate that all classes are of reasonable, non-trivial size and are interpretable. The sentence outlining this has been reworded for clarity and now reads; "The number of participants per class was substantial with the best fitting model classifying reasonable numbers (n=25/n=36/n=43) into categories that were readily interpretable (as showing consistent non-adherence; adherence which dropped off over time; and consistent adherence respectively). The three class solution was therefore selected for further study."
R2 (TV) 6	Reference 31 states "In preparation". The editor will have to decide whether that is acceptable.	The main trial paper is now in submission so this reference has been updated accordingly.
R3 (EB) 1	This paper described and tried to explain adherence to TSSE in patients with melanoma. Although interesting, this paper needs some improvements:	Thank you. Your suggested improvements are addressed below.
R3 (EB) 2	Please report in Methods section a power and sample size analysis. I'm afraid that some results strongly depend on the power of the study.	The sample size for this study was dictated by the sample size in the main ASICA trial (n=240; 120 in the intervention group). The ASICA trial was a pilot feasibility trial, and as such did not require a formal sample size calculation (Teare et al, 2014; Trials 2014;15:264). Indeed, one outcome of the feasibility trial was to provide the estimate effect sizes required to conduct a formal power calculation for the subsequent definitive trial. The final sample size of n=240 was a pragmatic choice based on high variability in HADS scores in a pilot

		sample of this population (and the need to recruit enough participants to capture this diversity) and practical estimates of the number of patients who could be recruited in each trial site in the time available. As the present analyses rely on the availability of objective records of TSSE (which were only collected in the ASICA intervention group), the present study used all of the available data from this group. We have amended the participants section to make clear that the sample size was pragmatically determined; “Sample size for the ASICA trial (n=240 total; n=120 intervention group) was pragmatically determined based on the need to capture anticipated diversity in the primary outcome (anxiety, depression, worry and quality of life) and the number of participants it was possible to recruit in the available time (28).”
R3 (EB) 3	Please report in Results section the ORs of "Intention" and "HADS depression" mutually adjusted and comment the results.	Multivariate analyses presenting the mutually adjusted estimates are now reported in a supplementary file. While we agree that it is important to look at mutually adjusted estimates, we have not included them in the main manuscript as we have concerns about whether it is appropriate to include self-efficacy and action planning concurrently in the same model. Efficacy is a theoretical precursor of action planning and so the two would be expected to be highly inter-related. The additional results, along with an accompanying commentary outlining the aforementioned caveat are outlined in the attached file S1. A line has been added to the main manuscript to direct readers to this file.
R3 (EB) 4	The Introduction section is too long and many parts are not informative with regards to the aim of the study.	The introduction has been reduced from 866 to 696 words. Introductory material on general melanoma has been removed to get to the core issue of recurrence more quickly, sections have been rephrased for brevity and the research questions have been condensed.

R3 (EB) 5	The "predictors of adherence" paragraph in Discussion section should be shortened in order to avoid redundancy.	The content of this paragraph and the two preceding paragraphs (initial summary of results and the 'Overall Adherence' section) have been reworked and partially combined to remove redundancy and repetition.
R3 (EB) 6	I am not sure about usefulness of Figure 2; it may be replaced by a sentence in Results section describing the overall observed pattern.	Figure 2 has been removed from the manuscript as suggested. See also the response to Reviewer 1 Point 8.

VERSION 2 – REVIEW

REVIEWER	Heckman, Carolyn Rutgers Cancer Institute of New Jersey, Division of Medicine
REVIEW RETURNED	31-May-2022

GENERAL COMMENTS	Thank you for the revisions. My one remaining concern is about the phrasing of the second research question. The study does not determine "why" there are differences in adherence. The second part of the question is more accurate. For the first part, you could say "what are the correlates of adherence" or "what variables predict adherence" or something along those lines, or just leave that part out entirely.
--

REVIEWER	Bidoli, Ettore IRCCS CRO AVIANO, Epidemiology
REVIEW RETURNED	27-May-2022

GENERAL COMMENTS	The paper can be accepted in the present form.
--

VERSION 2 – AUTHOR RESPONSE

Thank you for your feedback. The final comment to be addressed was;
Reviewer 1:” My one remaining concern is about the phrasing of the second research question. The study does not determine "why" there are differences in adherence. The second part of the question is more accurate. For the first part, you could say "what are the correlates of adherence" or "what variables predict adherence" or something along those lines, or just leave that part out entirely”.

To address this comment, we have amended the wording of the second research question as suggested. It now reads;

“What factors predict suboptimal TSSE adherence? That is, do groups with suboptimal adherence differ from those with optimal adherence on demographic (age, gender, socioeconomic status, rurality), socio-cognitive (intentions, self-efficacy, action plans) and/or emotional (recurrence worries, anxiety, depression) factors?”

As requested, two copies of the amended manuscript have been uploaded- the “Main Document R2” (clean copy) and the “Main Document R2 - marked copy” (changes highlighted). We have also made

some minor changes requested by the editorial assistant (referring to specific supplementary tables in the main text and re-ordering the statements at the end of the manuscript).